# Hydrogenation of β-Keto Sulfones to β-Hydroxy Sulfones with Alkyl Aluminum Compounds: Structure of Intermediate Hydroalumination Products

**DOI:** 10.3390/molecules27072357

**Published:** 2022-04-06

**Authors:** Michał Kotecki, Zbigniew Ochal, Paweł Socha, Vadim Szejko, Łukasz Dobrzycki, Mariola Stypik, Wanda Ziemkowska

**Affiliations:** 1Faculty of Chemistry, Warsaw University of Technology, Noakowskiego 3, 00-664 Warsaw, Poland; michal.kotecki2.stud@pw.edu.pl (M.K.); zbigniew.ochal@pw.edu.pl (Z.O.); vsheiko@ch.pw.edu.pl (V.S.); mariola.stypik.dokt@pw.edu.pl (M.S.); 2Department of Chemistry, University of Warsaw, Pasteura 1, 02-093 Warsaw, Poland; psocha@chem.uw.edu.pl (P.S.); dobrzyc@chem.uw.edu.pl (Ł.D.)

**Keywords:** β-keto sulfones, β-hydroxy sulfones, hydroalumination, hydrogenation, alkyl aluminum compounds

## Abstract

β-Hydroxy sulfones are important in organic synthesis. The simplest method of β-hydroxy sulfones synthesis is the hydrogenation of β-keto sulfones. Herein, we report the reducing properties of alkyl aluminum compounds R_3_Al (R = Et, *i*-Bu, *n*-Bu, *t*-Bu and *n*-Hex); *i*-Bu_2_AlH; Et_2_AlCl and EtAlCl_2_ in the hydrogenation of β-keto sulfones. The compounds *i*-Bu_2_AlH, *i*-Bu_3_Al and Et_3_Al are the at best reducing agents of β-keto sulfones to β-hydroxy sulfones. In reactions of β-keto sulfones with aluminum trialkyls, hydroalumination products with β-hydroxy sulfone ligands [R_2_AlOC(C_6_H_5_)CH_2_S(O)_2_(*p*-R^1^C_6_H_4_]_n_ [where *n* = 1,2; **2aa**: R = *i*-Bu, R^1^ = CH_3_; **2ab**: R = *i*-Bu, R^1^ = Cl; **2ba**: R = Et, R^1^ = CH_3_; **2bb**: R = Et, R^1^ = Cl] and {[Et_2_AlOC(C_6_H_5_)CH_2_S(O)_2_(*p*-ClC_6_H_4_]∙Et_3_Al}_n_
**3bb** were obtained. These complexes in the solid state have a dimeric structure, while in solutions, they appear as equilibrium monomer–dimer mixtures. The hydrolysis of both the isolated **2aa**, **2ab**, **2ba**, **2bb** and **3bb** and the postreaction mixtures quantitatively leads to pure racemic β-hydroxy sulfones. Hydroalumination reaction of β-keto sulfones with alkyl aluminum compounds and subsequent hydrolysis of the complexes is a simple and very efficient method of β-hydroxy sulfones synthesis.

## 1. Introduction

β-Hydroxy sulfones are motifs for the synthesis of a wide variety of organic products. The anions of these versatile β-hydroxy sulfones react, forming olefins by reductive elimination [1,2,3,4], vinyl sulfones by β-elimination reaction [5,6], lactones [7,8] and 2,5-disubstituted tetrahydrofurans [9,10]. It should be noted that chiral β-hydroxy sulfones are extremely useful building blocks for the synthesis of a variety of chiral organic compounds, e.g., γ-butenolides or allylic alcohols [11,12,13]. A number of methods for the β-hydroxy sulfones syntheses have been reported. They can be obtained, for instance, through a regioselective opening of β-epoxy sulfones [14] and oxiranes with various catalytic systems [15,16]. However, the reduction of carbonyl group of β-keto sulfones is considered as the most popular method of β-hydroxy sulfones synthesis. The reduction with NaBH_4_ without the addition of chiral additives leads to a racemic mixture of β-hydroxy sulfones [17,18,19,20], while enzymatic reduction and chemical enantioselective reduction of the C=O group lead to the chiral β-hydroxy sulfones with high enantioselectivity [21,22,23,24,25].

In the solution of β-keto sulfones, a tautomeric equilibrium takes place that is, however, almost completely shifted towards the ketone form (Figure 1).

Recently, we have found that the reaction between β-keto sulfones and *t*-Bu_2_AlH leads to the formation of aluminum complexes with β-hydroxy sulfone ligands, which indicates the reduction of β-keto sulfone to β-hydroxy sulfone by the alkyl aluminum compound [26]. The results of these studies inspired the development of a method for the β-hydroxy sulfones synthesis that uses aluminum alkyls bearing hydrogen atoms in the β-position of the alkyl substituents as β-keto sulfone-reducing agents.

It should be noted that, for many decades, alkyl aluminum compounds have been widely used in carbonyls reduction [27,28,29,30,31,32,33,34] and alkenes and alkynes hydroalumination reactions [35,36,37]. *i*-Bu_2_AlH is commonly used in selective reduction reactions, such as the reduction of trioxohexaaza[3.3.3]propelane to saturated hexaazapropelane derivatives, regioselective transformation of the CN group to the amine or the direct reduction of carboxylic acid esters to aldehydes [38,39,40].

In this paper, a β-keto sulfone reduction by various alkyl aluminum compounds, followed by the hydrolysis of the obtained aluminum complexes to β-hydroxy sulfones, is presented. Despite many methods that have been previously developed for the synthesis of chiral β-hydroxy sulfones, simple and efficient methods for the synthesis of racemic derivatives are still missing. We found that the efficiency of the reduction of β-keto sulfones to β-hydroxy sulfones depends mostly on the type of aluminum compounds, while the structure of β-keto sulfones affects the reduction process and the efficiency of β-hydroxy sulfone production to a lesser extent. Reactions of β-keto sulfones with *i*-Bu_3_Al and Et_3_Al, followed by the hydrolysis of postreaction mixtures, appear as a simple, efficient and cheap method of synthesizing β-hydroxy sulfones from starting β-keto sulfones. During the reaction of β-keto sulfones with aluminum alkyl compounds, complexes of aluminum alkyls with β-hydroxy sulfones as hydroalumination products are formed. The crystalline complexes were isolated and characterized by X-ray.

## 2. Results and Discussion

### 2.1. Hydroalumination Reaction of β-Keto Sulfones

β-Keto sulfones **1a**–**1e** were subjected to the reaction with alkyl aluminum compounds (*i*-Bu_3_Al, *i*-Bu_2_AlH, Et_3_Al, *n*-Bu_3_Al, *n*-Hex_3_Al, Et_2_AlCl and EtAlCl_2_), providing postreaction mixtures of β-keto sulfone hydroalumination products and the appropriate alkyl aluminum complex supported by β-keto sulfones. The compositions of the mixtures depended on the type of alkyl aluminum compounds and their reducing ability, as well as the structure of β-keto sulfones or the reaction conditions. The five hydroalumination products **2aa**, **2ab**, **2ba**, **2bb** and **3bb** were isolated as crystalline solids, and their structures were examined in the solid state (Figure 2). Moreover, all postreaction mixtures were subjected to hydrolysis in order to determine the degree of conversion of β-keto sulfones to β-hydroxy sulfones.

The treatment of 2-((4-methylphenyl)sulfonyl)-1-phenylethanol (**1a**) or 2-((4-chlorophenyl)sulfonyl)-1-phenylethanol (**1b**) with the one equivalent of *i*-Bu_3_Al or *i*-Bu_2_AlH in CH_2_Cl_2_, followed by crystallization from *n*-C_6_H_14_/CH_2_Cl_2_ solutions, afforded the crystalline β-keto sulfone hydroalumination products **2aa** and **2ab** (Figure 2). Reactions of **1a** and **1b** with Et_3_Al in a molar ratio of 1:1 led to the hydroalumination products **2ba** and **2bb**. When the β-keto sulfones:Et_3_Al molar ratio was changed to 1:2, in the obtained compounds, an additional Et_3_Al molecule was coordinated to SO_2_ oxygen atoms. Compound **3bb** was crystallized and characterized (Figure 2).

The molecular structures of compounds **2aa**, **2ab**, **2ba**, **2bb** and **3bb** were determined by X-ray diffraction study and are shown in Figure 1, Figure 2, Figure 3, Figure 4 and Figure 5. Data collection and structure analyses are listed in Appendix A. In the solid state, all of the described compounds were presented as centrosymmetric dimers. They consisted of central four-membered Al_2_O_2_ rings formed by two monoanionic β-hydroxy sulfonic ligands and two alkylaluminium moieties with four-coordinate aluminum centrum. Additionally, in the **3bb** molecule, there were two Et_3_Al molecules coordinated to the oxygen atoms in the SO_2_ groups. The sum of the angles around the O(3) atoms was 354.9° for compound **2aa** and 354.7° for compound **2ab**, which indicated slight stress in the central part of the molecule. Similarly, the sums of the angles around the oxygen atoms of the Al_2_O_2_ rings in compounds **2ba**, **2bb** and **3bb** were 354.8, 354.7 and 355.6°, respectively.

The central Al_2_O_2_ rings are similar to that of typical alkoxides of group 13 metal alkyls obtained in reactions of R_3_M (R = Me, Et, *i*-Bu; M = Al, Ga) with diverse monoalcohols [41,42,43]. The bond lengths C(1)-C(2) [1.533(2) Å in **2aa**] C(7)-C(8) [1.533(2) Å in **2ab**], C(1)-C(8) [1.534(2) Å in **2ba**], C(5)-C(12) [1.538(2) Å in **2bb**] and C(6)-C(13) [1.533(4) Å in **3bb**] are typical for single C–C bonds, which proves the transformation of the C=C double bonds in the β-keto sulfones into single C–C bonds in the appropriate β-hydroxy sulfone residues.

Surprisingly, on the basis of NMR spectra of compounds **2aa**, **2ab**, **2ba**, **2bb** and **3bb**, it was found that there are two types of structures in the solutions. Such was observed for both redissolved crystalline solids, as well as for postreaction mixtures. This was evidenced by the presence of four signals deriving from the alkyl groups of the alkyl aluminum moieties. For compound **2aa**, four overlapping doublets at 0.81, 0.77, 0.76 and 0.71 ppm of AlCH_2_C(H)(CH_3_)_2_) protons and four doublets at −0.28, −0.39, −0.41 and −0.51 ppm of AlCH_2_C(H)(CH_3_)_2_) protons were observed (Appendix A). Similarly, in the ^1^H NMR spectrum of compound **2ab,** the following signals of *i*-Bu protons were present: four overlapping doublets at 0.82, 0.78, 0.77 and 0.72 ppm of AlCH_2_C(H)(CH_3_)_2_) protons and four doublets at −0.26, −0.37, −0.39 and −0.50 ppm of AlCH_2_C(H)(CH_3_)_2_) protons (Appendix A). For compound **2ba**, one triplet at 0.80 ppm, two overlapping triplets at 0.72 ppm and one triplet at 0.64 ppm of AlCH_2_CH_3_ protons were observed, whereas the signals of AlCH_2_CH_3_ protons appeared as four quartets at −0.39, −0.53 (two overlapping signals) and −0.65 ppm. Signals of two structures of **2bb** were also observed in the ^1^H NMR spectrum: at 0.81, 0.73 (two overlapping triplets) and 0.65 ppm triplets of AlCH_2_CH_3_ protons and four quartets at −0.36, −0.50, −0.51 and −0.64 of AlCH_2_CH_3_ protons.

In compound **3bb**, due to the presence of Et_3_Al molecules coordinated to the oxygen atoms from the SO_2_ groups, there was an additional triplet of (CH_3_CH_2_)_3_Al protons and a quartet of (CH_3_CH_2_)_3_Al protons (at 0.92 and −0.29 ppm, respectively) in the ^1^H NMR spectrum (Appendix A). In addition, there were four triplets at 1.03, 0.84, 0.74 and 0.63 ppm of CH_3_CH_2_Al protons; three quartets at −0.03, −0.46, −0.64 ppm and one quartet at −0.29 ppm overlapping the signal of the (CH_3_CH_2_)_3_Al protons.

The ^13^C NMR spectra of the compounds revealed two signals of (SCH_2_CH) carbon atoms (at 71.86 and 71.81 ppm for **2aa**, at 72.09 and 72.04 ppm for **2ab**, at 71.42 and 71.40 ppm for **2bb**, at 71.02 and 70.95 ppm for **3bb** and at 71.40 ppm broadened for **2ba**), which also confirmed the presence of two structures in solutions. Likewise, instead of single signals, the SCH_2_CH carbon atoms showed two signals: at 61.92 and 61.85 ppm for **2aa**, at 62.17 and 62.09 ppm for **2ab**, at 61.66 and 61.64 ppm for **2ba**, at 61.59 ppm broadened for **2bb and** at 61.74 and 61.45 ppm for **3bb**.

The complex nature of the NMR spectra of **2aa**, **2ab**, **2ba**, **2bb** and **3bb** complexes could be explained by the monomer–dimer equilibria in the solutions (Figure 3). To confirm this, the molecular weight of the dissolved compounds was determined by the cryometric method. In the solid state, the compounds had the structures of dimeric (R*,S*) diastereomers, as shown by X-ray measurements (Figure 1, Figure 2, Figure 3, Figure 4 and Figure 5). After dissolving the compounds, Al_2_O_2_ rings in dimeric structures were easily dissociated to form monomeric structures stabilized by the formation of Al–O coordination bonds between the oxygen atoms of the SO_2_ group and aluminum atoms. The association degrees calculated from the values of molecular weights ranged from 1.22 (for **3bb**) to 1.50 (for **2ab**), which means that, in solutions of compounds **3bb** and **2ab**, there were 22 and 49 mol% of the dimeric structure, respectively. Taking into account the results of NMR studies and molecular weight measurements, it can be concluded that hydroalumination products of β-keto sulfones exist as an equilibrium mixture of monomers–dimers in solutions (Figure 3).

Since the tautomeric equilibrium in the β-keto sulfones solutions was almost completely shifted towards the ketone form, only this form was taken into account in the hydroalumination mechanism suggested. When *i*-Bu_2_AlH was used, the mechanism was based on the assumption of a charge distribution between the carbonyl C=O and Al-H groups, allowing the formation of an intermediate state. The oxygen atom in the C=O group with a partially negative charge interacted with a partially positive aluminum, and the partially negative charged hydrogen atom Al–H was transferred to the C=O carbon atom simultaneously (Figure 4). We have recently proposed a similar mechanism for the hydroalumination of β-keto sulfones with *t*-Bu_2_AlH [26].

In the reactions of β-keto sulfones with *i*-Bu_3_Al an Et_3_Al, β-hydrogen from the *i*-Bu or Et group bonded to the partially positive C=O carbon, and the aluminum atom interacted with the negative oxygen atom C=O. An intermediate state involving six atoms, AlCCHCO, was formed. In the next step, the alkene molecule was removed, and the aluminum complex of β-hydroxy sulfone was formed (Figure 4). The similar mechanism was previously published by Ashby for a ketone reduction reaction with *i*-Bu_3_Al [27].

### 2.2. Hydrogenation of β-Keto Sulfones to β-Hydroxy Sulfones

Reaction mixtures of β-keto sulfones with aluminum compounds were hydrolyzed to decompose the complexes. The obtained products were characterized by NMR spectroscopy to determine the molar ratio of β-hydroxy sulfone to β-keto sulfone on the basis of an integration of SO_2_CH proton signals in β-hydroxy sulfone and in β-keto sulfone. The yield of β-hydroxy sulfones (Table 1) illustrated an efficiency of the β-keto sulfone hydrogenation process. We determined the effect of the structure of β-keto sulfones, the type of aluminum compound and the reaction conditions on the efficiency of the hydrogenation of β-keto sulfones to β-hydroxy sulfones. We found that the hydrogenation reaction depended primarily on the nature of the aluminum alkyl compound. The most active reagent was *i*-Bu_3_Al, which reduced quantitatively all β-keto sulfones regardless of their structure. Et_3_Al was a good reducer for β-keto sulfones **1a**,**b** and **1d**,**e**, with electron-withdrawing substituents in the β-position, while the hydrogenation of β-keto sulfone **1c** with an electron-donating methyl group was 75% efficient. Using an excess of Et_3_Al slightly increased the yield of β-hydroxy sulfone **4c** to 82% (Table 1, run 3). The activity of *n*-Bu_3_Al, *n*-Hex_3_Al and *t*-Bu_3_Al in the hydrogenation of β-keto sulfones was weaker compared to the activity of *i*-Bu_3_Al and Et_3_Al. However, using an excess of *n*-Hex_3_Al and *t*-Bu_3_Al to reduce the β-keto sulfones **1a** and **1b** resulted in a significant increase in yield from 55 to 100% and from 8 to 92%, respectively (Table 1, runs 1 and 2). The presence of chloride substituents in alkyl aluminum compounds significantly reduced the activity of these compounds in the hydrogenation of β-hydroxy sulfones. In the presence of an equimolar amount of Et_2_AlCl only 17% of the beta keto sulfone, **1b** was reduced. For a 1:2 molar ratio of Et_2_AlCl:**1b**, β-hydroxy sulfone **4b** was obtained with a yield of 25% (Table 1, run 2). EtAlCl_2_ was inactive in the hydrogenation of β-keto sulfones (Table 1, run 1).

The nature of the starting β-keto sulfones had a less significant effect on their ability to be hydrogenated with alkyl aluminum compounds. The presence of electron-withdrawing groups on the C=O carbon atom, such as the phenyl substituent in compounds **1a**,**b** and **1d**,**e**, caused an increase in the partial positive charge on the C=O carbon atom, which favored the reduction of β-keto sulfones, as shown in the Figure 4.

Earlier studies on ketone hydrogenation showed that the presence of a Lewis base (e.g., diethyl ether, THF) inactivates the reducing properties of aluminum alkyls [31]. That was why we used methylene dichloride, *n*-pentane and *n*-hexane as solvents; however, methylene dichloride proved to be the best due to the good solubility of the compounds.

The reaction of aluminum alkyls with β-keto sulfones and subsequent hydrolysis of postreaction mixtures was a simple method of β-keto sulfones hydrogenation. However, this method was suitable when the β-keto sulfone was completely hydroaluminated by an alkyl aluminum compound. On the other hand, in the presence of less active aluminum alkyls, only a part of the β-keto sulfone could be hydroaluminated. Then, in the postreaction mixture, there were alkyl aluminum complexes with β-hydroxy sulfone and β-keto sulfone ligands, which, after hydrolysis, yielded a mixture of β-hydroxy sulfone and β-keto sulfone. In order to avoid a difficult separation of β-hydroxy sulfone from this mixture, the alkyl aluminum complex with β-hydroxy sulfone ligands should be crystallized from the reaction mixture and then hydrolyzed to pure β-hydroxy sulfone. Complexes with β-keto sulfone ligands were thick liquids, which facilitated the separation of solid complexes with β-hydroxy sulfone ligands.

## 3. Materials and Methods

### 3.1. General Remarks

All manipulations were carried out using standard Schlenk techniques under an inert gas atmosphere. Methylene dichloride was deacidified with basic Al_2_O_3_ and distilled over P_2_O_5_ under argon. ^1^H and ^13^C NMR spectra were obtained on a Varian Mercury-400 MHz spectrometer (Varian International AG, Switzerland). Chemical shifts were referenced to the residual proton signals of CDCl_3_ (7.26 ppm). ^13^C NMR spectra were acquired at 100.60 MHz (standard: chloroform ^13^CDCl_3_, 77.20 ppm). NMR spectra can be found in the Appendix A. Tri-*iso*-butyl aluminum and di-*iso*-butyl aluminum hydride were from Sigma-Aldrich Company (Poznań, Poland). β-Keto sulfones **1a**–**e** were synthesized according to the literature data [44]. Hydrolysable alkyl groups bonded to Al atoms for products **2aa**, **2ab**, **2ba**, **2bb** and **3bb** were determined by hydrolysis of the compound (0.2 to 0.3 g) using HNO_3_ solution (10% concentrated, 5 cm^3^) and measurement of the volume of gaseous alkane (C_4_H_10_ or C_2_H_6_). Subsequently, the sample was transformed into Al_2_O_3_ by mineralization, and the obtained white solid was dissolved in a diluted water solution of HNO_3_. The content of aluminum was determined by the complexation of Al^3+^ cations with versenate anions using an excess of the titrated solution of calcium disodium versenate. Then, the excess of calcium disodium versenate was titrated by FeCl_3_.

### 3.2. X-ray Crystallography

The X-ray measurements of compounds **2aa**, **2ab**, **2ba**, **2bb** and **3bb** were performed at 100(2) K on a Bruker D8 Venture Photon100 diffractometer equipped with a TRIUMPH monochromator and a MoKα fine focus-sealed tube (λ = 0.71073 Å). The total frames were collected with the Bruker APEX2 program [45]. The temperature of the samples was 100 K. The frames were integrated with the Bruker SAINT software package [46] using a narrow frame algorithm. Data were corrected for absorption effects using the multi-scan method (SADABS) [47]. The structures were solved and refined using the SHELXTL software package [48,49]. The atomic scattering factors were taken from the International Tables [50]. All hydrogen atoms were placed in calculated positions and refined within the riding model. Detailed crystallographic data are listed in Appendix A.

### 3.3. Reactions of β-Keto Sulfones with Alkyl Aluminum Compounds—General Procedure

A solution of a suitable amount of alkyl aluminum compound in methylene dichloride was added to a solution of 2 mmol of β-keto sulfone in 10 cm^3^ of methylene dichloride at 0–5 °C with stirring. After warming up to room temperature, the postreaction mixture was subjected to hydrolysis.

### 3.4. Preparation of Hydroalumination Products

#### Reactions of *i*-Bu_3_Al, *i*-Bu_2_AlH and Et_3_Al with β-Keto Sulfones

A solution of *i*-Bu_2_AlH (0.284 g, 2 mmol) or *i*-Bu_3_Al (0.396 g, 2 mmol) in 10 cm^3^ of methylene dichloride was added to a solution of β-keto sulfone (0.548 g, 2 mmol of **1a** or 0.589 g, 2 mmol of **1b**) in 10 cm^3^ at 0–5 °C with stirring. A solution of Et_3_Al (0.228 g, 2 mmol) in 10 cm^3^ of methylene dichloride was added to a solution of β-keto sulfone (0.548 g, 2 mmol of **1a** or 0.589 g or 2 mmol of **1b**) in 10 cm^3^ at −76 °C with stirring. A solution of Et_3_Al (0.456 g, 2 mmol) in 20 cm^3^ of methylene dichloride was added to a solution of β-keto sulfone **1b** (0.589 g, 2 mmol) in 10 cm^3^ at −76 °C with stirring. The mixtures were stirred for 1 h at this temperature and then allowed to warm to ambient temperature. The solvent was removed from the postreaction mixtures by distillation under vacuum. A thick liquid was obtained when the reagent was *i*-Bu_2_AlH, while white solids were obtained when the reagents were *i*-Bu_3_Al and Et_3_Al. White crystals of the complexes **2aa**, **2ab**, **2ba**, **2bb** and **3bb** suitable for X-ray measurements were precipitated from *n*-C_6_H_14_/CH_2_Cl_2_ solutions. Before measuring the molecular weight by the cryoscopic method in benzene and analysis, samples of compounds were placed under vacuum (10^−2^ Torr) for 5 h to remove the solvent. Yield: *i*-Bu_3_Al reacted with β-keto sulfones **1a** and **1b**, yielding compounds **2aa** and **2ab** quantitatively (based on NMR spectra), while postreaction mixtures of *i*-Bu_2_AlH with β-keto sulfones **1a** and **1b**, besides **2aa** and **2ab**, consisted of side products.

*Di-iso-butyl aluminum complex with 2-((4-methylphenyl)sulfonyl)-1-phenylethanol* (**2aa**): ^1^H NMR (Appendix A) δ: 7.40–7.15 (9H, m, H_aromat_), 5.20 (1H, m, CH), 3.93–3.75 (2H, m, CH_2_), 2.37 (3H, s, CH_3_), 1.49, 1.40 and 1.31 (2 H, 3 multiplets, AlCH_2_C(H)(CH_3_)_2_), 0.81, 0.77, 0.76 and 0.71 (6H, 4 overlapping doublets, ^3^J_H_—4 Hz, AlCH_2_C(H)(CH_3_)_2_), −0.28, −0.39, −0.41 and −0.51 (4H, 4 doublets, ^3^J_H_—4 Hz, AlCH_2_C(H)(CH_3_)_2_). ^13^C NMR (Appendix A) δ 144.69, 144.65, 136.53, 136.45, 135.90, 135.86 129.80, 129.66, 129.65, 128.98, 128.38, 127.75, 127.72 (C_aromat_), 71.86, 71.81 (SCH_2_CH), 61.92, 61.85 (SCH_2_CH), 28.13, 28,10, 28.03 (AlCH_2_C(H)(CH_3_)_2_), 25.37, 25.27, 25.18 (AlCH_2_C(H)(CH_3_)_2_), 23.03, 22.99, 22.82 (AlCH_2_C(H)(CH_3_)_2_), 21.52 (PhCH_3_) ppm. Mp.: 153–156 °C. Molecular weight: 590 g/mol (cal. for **2aa** monomer 416.5 g/mol; for **2aa** dimer 833 g/mol). Anal. Al, 6.15; hydrolysable *i*-Bu groups, 26.55; calcd for **2aa** (C_46_H_66_Al_2_O_6_S_2_): Al, 6.49; *i*-Bu groups, 27.40 wt%.

*Di-iso-butyl aluminum complex with 2-((4-chlorophenyl)sulfonyl)-1-phenylethanol* (**2ab**): ^1^H NMR (Appendix A) δ: 7.38–7.13 (9H, m, H_aromat_), 5.24 (1H, m, CH), 3.93–3.77 (2H, m, CH_2_), 1.50, 1.40 and 1.32 (2 H, 3 multiplets, AlCH_2_C(H)(CH_3_)_2_), 0.82, 0.78, 0.77 and 0.72 (6H, 4 overlapping doublets, ^3^J_H_—4 Hz, AlCH_2_C(H)(CH_3_)_2_), −0.26, −0.37, −0.39 and −0.50 (4H, 4 doublets, ^3^J_H_—4 Hz, AlCH_2_C(H)(CH_3_)_2_). ^13^C NMR (Appendix A) δ: 140.63, 140.60, 137.43, 137.38, 136.31, 136.23, 130.35, 129.52, 129.38, 129.35, 128.65 (C_aromat_), 72.09, 72.04 (CH_2_CH), 62.17, 62.09 (S-CH_2_), 28.40, 28,37, 28.31 (AlCH_2_C(H)(CH_3_)_2_), 25.67, 25.57, 25.49 (AlCH_2_C(H)(CH_3_)_2_), 23.33, 23.26, 23.04 (broad, AlCH_2_C(H)(CH_3_)_2_) ppm. Mp.: 113–118 °C. Molecular weight: 651 g/mol (calc. for **2ab** monomer 436.5 g/mol; for **2ab** dimer 873 g/mol). Anal. Al, 5.87; hydrolysable *i*-Bu groups, 25.30; calcd for **2ab** (C_44_H_60_Al_2_Cl_2_O_6_S_2_): Al, 6.18; *i*-Bu groups, 26.09 wt%.

*Di-ethyl aluminum complex with 2-((4-methylphenyl)sulfonyl)-1-phenylethanol* (**2ba**): ^1^H NMR (Appendix A) δ: 7.42 (2H, m, H_aromat_), 7.28–7.11 (7H, m, H_aromat_), 5.17 (1H, m, CH), 3.88–3.80 (1H, m, CH_2_), 3.73–3.67 (1H, m, CH_2_), 2.35 (3H, s, CH_3_Ph), 0.80 (1.5H, t, AlCH_2_CH_3_), 0.72 (3H, two overlapping triplets, AlCH_2_CH_3_), 0.64 (1.5H, t, AlCH_2_CH_3_), −0.39 (1H, q, AlCH_2_CH_3_), −0.53, −0.53 (2H, two quartets, AlCH_2_CH_3_), −0.65 (1H, q, AlCH_2_CH_3_). ^13^C NMR (Appendix A) δ: 144.80, 144.77, 136.90, 135.87, 135.60, 129.73, 129.69, 129.67, 128.93, 127.86, 127.75, 127.74 (Caromat), 71.40 (CH_2_CH, broadened), 61.66, 61.64 (S-CH_2_), 21.53 (CH_3_Ph), 8.63, 8.57, 8.49 (AlCH_2_CH3), 0.42 (AlCH_2_CH_3_, broadened) ppm. Mp.: = 132–136 °C. Molecular weight: 450 g/mol (calc. for **2ba** monomer 360 g/mol; for **2ba** dimer 720 g/mol). Anal. Al, 7.28; hydrolysable Et groups, 15.82; calcd for **2ba** (C_38_H_50_Al_2_O_6_S_2_): Al, 7.50; Et groups, 16.11 wt%.

*Di-ethyl aluminum complex with 2-((4-chlorophenyl)sulfonyl)-1-phenylethanol* (**2bb**): ^1^H NMR (Appendix A) δ: 7.40–7.12 (9H, m, H_aromat_), 5.19 (1H, m, CH), 3.90–3.69 (2H, m, CH_2_), 0.81 (1.5H, t, AlCH_2_CH_3_), 0.73 (3H, two triplets, AlCH_2_CH_3_), 0.65 (1.5H, t, AlCH_2_CH_3_), −0.36 (1H, q, AlCH_2_CH_3_), −0.50, −0.51 (2H, two quartets, AlCH_2_CH_3_), −0.64 (1H, q, AlCH_2_CH_3_). ^13^C NMR (Appendix A) δ: 140.43, 140.40, 136.92, 136.34, 130.02, 129.28, 129.26, 129.16, 129.14, 129.09, 127.86 (C_aromat_), 71.42, 71.40 (CH_2_CH), 61.59 (S-CH_2_), 8.62, 8.56, 8.48 (AlCH_2_CH_3_), 0.35 (AlCH_2_CH_3_, broadened) ppm. Mp.: 130–133 °C. Molecular weight: 505 g/mol (calc. for **2bb** monomer 380.5 g/mol; for **2bb** dimer 761 g/mol). Anal. Al, 7.01; hydrolysable Et groups, 15.79; calcd for **2bb** (C_36_H_44_Al_2_Cl_2_O_6_S_2_): Al, 7.10; Et groups, 16.11 wt%.

*Di-ethyl aluminum complex with 2-((4-chlorophenyl)sulfonyl)-1-phenylethanol and triethyl aluminum* (**3bb**): ^1^H NMR (Appendix A) δ: 7.31–7.08 (9H, m, H_aromat_), 5.20 (1H, m, CH), 4.26–3.94 (2H, m, CH_2_), 1.03, 0.84, 0.74, 0.63 (6H, four triplets, AlCH_2_CH_3_), 0.92 (9H, t, Al(CH_2_CH_3_)_3_), −0.03, −0.46, −0.64 (3H, 3q, AlCH_2_CH_3_), −0.29 (6H of Al(CH_2_CH_3_)_3_ and 1H of AlCH_2_CH_3_, q, AlCH_2_CH_3_). ^13^C NMR (Appendix A) δ: 142.50, 142.47, 134.90, 134.83, 133.28, 130.79, 129.99, 129.91, 129.56, 129.08, 129.05, 128.72, 128.49, 128.35, 127.77, 125.39 (C_aromat_), 71.02, 70.95 (CH_2_CH), 61.74, 61.45 (S-CH_2_), 9.38, 8.50, 8.40, 8.27 (AlCH_2_CH_3_), 1.05, 0.67, 0.16, 0.08 (AlCH_2_CH_3_) ppm. Mp.: 148–150 °C. Molecular weight: 604 g/mol (calc. for **3bb** monomer 494.5 g/mol; for **3bb** dimer 989 g/mol). Anal. Al, 10.65; hydrolysable Et groups, 28.97; calcd for **3bb** (C_48_H_74_Al_4_Cl_2_O_6_S_2)_: Al, 10.92; Et groups, 29.32 wt%.

### 3.5. Preparation of β-Hydroxy Sulfones

Method 1: Hydrolysis of isolated compounds **2aa**, **2ab**, **2ba**, **2bb** and **3bb**. A solution of 0.5 mmol of compounds **2** (or **3bb**) in 10 cm^3^ of CH_2_Cl_2_ and 10 cm^3^ of a 10% solution of hydrochloric acid was added to the separating funnel. After shaking, the organic layer was separated, and the aqueous layer was washed twice with 10 cm^3^ of CH_2_Cl_2_. The organic layers were combined, and the solvent was distilled under vacuum. White solids of a pure β-hydroxy sulfones **4a** (or **4b**) were obtained.

Method 2: Hydrolysis of postreaction mixtures of the reactions of β-keto sulfones **1a**–**1e** with aluminum compounds. The postreaction mixtures of the reaction of 0.5 mmol of β-keto sulfone (10 cm^3^ of the CH_2_Cl_2_ solution) reacted with water, according to the procedure described in Method 1.

The results of the conversion of β-keto sulfones to β-hydroxy sulfones are presented in Table 1. Mp of 2-hydroxy-2-phenyethyl-4-mehylphenylsulfone **4a**: 74–75 °C, (literature data 69–71 °C [17], 69.4–70.8 °C [51], 78–79 °C [52] and 74–75 °C [53]; Mp of 2-hydroxy-2-phenyethyl-4-chlorophenylsulfone **4b**: 105–107 °C (literature data 106–108 °C [53], 105–106 °C [54] and 103.5–105 °C [55]); Mp of 1-(4-methylphenylsulfonyl)propan-2-ol **4c**: 75–76 °C (literature data 78 °C [56]); Mp of 2-[(4’-methylphenyl)sulfonyl]-1,2-diphenylethanol **4d**: 159–160 °C (literature data 156–157 °C [57]) and Mp of 1-phenyl-2-(4-methylphenylsulfonyl)propan-1-ol **4e**: 100–103 °C (literature data 99–100.5 °C [58]).

## 4. Conclusions

Although aluminum trialkyls R_3_Al with substituents that have β-hydrogens are active reducing agents of β-keto sulfones to β-hydroxy sulfones, the reducing properties of aluminum *iso*-butyl compounds (*i*-Bu_3_Al and *i*-Bu_2_AlH) and Et_3_Al are the greatest. In reactions of β-keto sulfones with R_3_Al, the hydroalumination of β-keto sulfones takes place, resulting in the formation of aluminum complexes with β-hydroxy sulfones considered as intermediates in the production of β-hydroxy sulfones. In the solid state, these complexes exhibit as dimers, while, in solutions, they undergo an equilibrium between monomeric and dimeric forms. The hydrolysis of both the isolated aluminum complexes with β-hydroxy sulfones and the postreaction mixtures quantitatively lead to pure racemic β-hydroxy sulfones. Summarizing, the hydroalumination reaction of β-keto sulfones with *i*-Bu_3_Al, *i*-Bu_2_AlH and Et_3_Al, followed by the hydrolysis of the resulting complexes in the postreaction mixtures, is a simple and efficient method for racemic β-hydroxy sulfones.

## Data Availability

The date presented in this study are available in the Appendix A.

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
