# Peer review of "Hydrogenation of β-Keto Sulfones to β-Hydroxy Sulfones with Alkyl Aluminum Compounds: Structure of Intermediate Hydroalumination Products"

_molecules, 2022, doi:10.3390/molecules27072357_

Round 1

Reviewer 1 Report

The article entitled “Hydrogenation of β-keto sulfones to β-hydroxy sulfones with 2 alkyl aluminum compounds. Structure of intermediate hydroalumination products” describes the reduction reaction of a series of β-keto sulfones with trialkylaluminum compounds. The reaction produces on hydrolysis the reduction of the carbonyl group.

According to my opinion the article requires a thorough review, and it is not publishable in the current form. The manuscript describes, indeed, a reaction already described starting from 1970 and applies it in the reduction of β-keto sulfones. The authors in the introduction point out the importance of chiral β-hydroxy sulfones but the methodology they described is racemic. So, according to my opinion, Authors need to rewrite the introduction to indicate why it is important that their work be published.

Here are some other points to correct:

  1. English must be correct in many sentences. See for example: from lines 14 to 16 “herein ……..sulfones”; from lines 16 to 18 “among ……..sulfones”; from lines 78 to 80 “the composition …..conditions”; from lines 178 to 180 “it should be….diastereomers”
  2. The Authors should explain what they intend for making a hydrolysis on an isolated compound or on “a post reaction mixture” (lines 24, 76, 82, 95 etc.…)
  3. The arrow in Scheme 1 is wrong. Conventionally this type of arrow is used to indicate resonance isomers and tautomers are not.
  4. Line 178. 1a and 1b are not β-hydroxy sulfones
  5. Schema 4. Use the delta symbol instead of sigma one.
  6. From lines 218 to 225. This part is a repetition of concepts that have already been said in the previous paragraphs.
  7. From lines 180 to 183 The Authors attribute the presence of diastereomers in the NMR spectrum as a result of an equilibrium between the monomer and the dimer. Why then include these considerations? They are useless and they burden the reasoning

Author Response

The article entitled “Hydrogenation of β-keto sulfones to β-hydroxy sulfones with 2 alkyl aluminum compounds. Structure of intermediate hydroalumination products” describes the reduction reaction of a series of β-keto sulfones with trialkylaluminum compounds. The reaction produces on hydrolysis the reduction of the carbonyl group.

According to my opinion the article requires a thorough review, and it is not publishable in the current form. The manuscript describes, indeed, a reaction already described starting from 1970 and applies it in the reduction of β-keto sulfones. The authors in the introduction point out the importance of chiral β-hydroxy sulfones but the methodology they described is racemic. So, according to my opinion, Authors need to rewrite the introduction to indicate why it is important that their work be published.

Answer: The authors thank the Reviewer for their insightful review. The Introduction has been rebuilt. New references have been added. At the end of the Introduction, the following sentence was added: “Although many methods have been developed for the synthesis of chiral β-hydroxy sul-fones, simple and efficient methods for the synthesis of racemic derivatives are needed.”

Here are some other points to correct:

  1. English must be correct in many sentences. See for example: from lines 14 to 16 “herein ……..sulfones”; from lines 16 to 18 “among ……..sulfones”; from lines 78 to 80 “the composition …..conditions”; from lines 178 to 180 “it should be….diastereomers”

Answer: The linguistic revision of the entire article was carried out.

  1. The Authors should explain what they intend for making a hydrolysis on an isolated compound or on “a post reaction mixture” (lines 24, 76, 82, 95 etc.…)

Answer: In the part 2.2. Hydrogenation of β-keto sulfones to β-hydroxy sulfones the following fragmentation was added: “The reaction of aluminum alkyls with β-keto sulfones and subsequent hydrolysis of post-reaction mixtures is a simple method of the β-keto sulfones hydrogenation. However, this method is suitable when the β-keto sulfone is completely hydroaluminated by alkyl aluminum compound. On the other hand, in the presence of less active aluminum alkyls, only a part of the β-keto sulfone could be hydroaluminated. Then, in the post-reaction mixture, there are alkyl aluminum complexes with β-hydroxy sulfone and β-keto sulfone ligands, which, after hydrolysis, yield a mixture of β-hydroxy sulfone and β-keto sulfone. In order to avoid difficult separation of β-hydroxy sulfone from this mixture, the alkyl aluminum complex with β-hydroxy sulfone ligands should be crystallized from the reaction mixture and then hydrolyzed to pure β-hydroxy sulfone. Complexes with β-keto sulfone ligands are thick liquids which facilitates the separation of solid complexes with β-hydroxy sulfone ligands.”

  1. The arrow in Scheme 1 is wrong. Conventionally this type of arrow is used to indicate resonance isomers and tautomers are not.

Answer: The arrow was changed.

  1. Line 178. 1a and 1b are not β-hydroxy sulfones

Answer: This part of text has been removed.

  1. Schema 4. Use the delta symbol instead of sigma one.

Answer: The delta symbol was used instead of sigma one.

  1. From lines 218 to 225. This part is a repetition of concepts that have already been said in the previous paragraphs.

Answer: This part was removed.

  1. From lines 180 to 183 The Authors attribute the presence of diastereomers in the NMR spectrum as a result of an equilibrium between the monomer and the dimer. Why then include these considerations? They are useless and they burden the reasoning.

Answer: The considerations were removed.

Reviewer 2 Report

An evaluation of different alkyl aluminum compounds as reducing agents of beta-keto sulfones is presented. Although the main synthetic applications of the resulting products, i.e. beta-hydroxy sulfones, is its use as chiral reagents in organic synthesis, obtaining these derivatives in racemic form in a simple and efficient way is not without interest. In this sense, the study presented herein is a good contribution to the field, carried out rigorously, and deserves to be published. Acceptance is recommended after addressing the following minor points:

  • Page 2 (line 83): sulfones instead of suflfones.
  • Please make attention to Figures 3 and 4 since the numbering given in the captions does not correspond to that of Scheme 2.
  • It would be interesting to know if the equilibrium mixtures present in solution (Scheme 2) are temperature dependent. Have the authors performed variable-temperature NMR experiments?

Author Response

An evaluation of different alkyl aluminum compounds as reducing agents of beta-keto sulfones is presented. Although the main synthetic applications of the resulting products, i.e. beta-hydroxy sulfones, is its use as chiral reagents in organic synthesis, obtaining these derivatives in racemic form in a simple and efficient way is not without interest. In this sense, the study presented herein is a good contribution to the field, carried out rigorously, and deserves to be published. Acceptance is recommended after addressing the following minor points:

Answer: The authors thank the Reviewer for their insightful review.

  • Page 2 (line 83): sulfones instead of suflfones.

Answer: The name “sulfones” was introduced.

  • Please make attention to Figures 3 and 4 since the numbering given in the captions does not correspond to that of Scheme 2.

Answer: the numbering was changed.

  • It would be interesting to know if the equilibrium mixtures present in solution (Scheme 2) are temperature dependent. Have the authors performed variable-temperature NMR experiments?

Answer: Low temperature NMR spectra are difficult to make because the compounds are poorly soluble. Even after a slight decrease in temperature, a precipitate begins to form in the NMR tube, which changes the dimer: monomer ratio in the solution.

Round 2

Reviewer 1 Report

The article entitled “Hydrogenation of β-keto sulfones to β-hydroxy sulfones with 2 alkyl aluminum compounds. Structure of intermediate hydroalumination products” is now, according to my opinion, accptable for publication in Molecules.